# Enhancing Foundation Models with Federated Domain Knowledge Infusion

Jiaqi Wang [1]   Jingtao Li [2]   Weiming Zhuang [2]   Chen Chen [2]   Lingjuan Lyu [2]   Fenglong Ma [1]

## Abstract

Vision foundation models (FMs) like CLIP have exhibited exceptional capabilities in visual and linguistic understanding, particularly in zero-shot inference tasks. However, these models struggle with data that significantly deviates from their training samples, necessitating fine-tuning, which is often infeasible in centralized settings due to data privacy concerns. Federated learning (FL) combined with parameter-efficient fine-tuning (PEFT) offers a potential solution, yet existing methods face issues with domain-specific characteristics and out-of-domain generalization. We propose a cross-silo Federated Adapter Generalization (`FedAG`), a novel federated fine-tuning approach that leverages multiple fine-grained adapters to capture domain-specific knowledge while enhancing out-of-domain generalization. Our method uses quality-aware in-domain mutual learning and attention-regularized cross-domain learning to integrate domain-specific insights effectively. Experiments of the CLIP model on three domain-shifting datasets, ImageCLEF-DA, Office-Home, and DomainNet, demonstrate the superior performance of `FedAG` in both in-domain and out-of-domain scenarios. We envision this work as a milestone for generalizing CLIP to handle the challenge of out-of-domain knowledge under federated learning setting. The source code can be found at https://github.com/JackqqWang/fedag.

## 1. Introduction

CLIP (Contrastive Language–Image Pre-training) (Radford et al., 2021) and its variants (Li et al., 2023), have demonstrated superior capabilities in understanding visual concepts and their linguistic descriptions. They have been employed in a wide range of vision tasks, including image classification, especially for zero-shot inference, thanks to their large number of parameters and the extensive training data. However, these models still face challenges when confronted with input data significantly different from their training data. Therefore, fine-tuning becomes essential. Traditional fine-tuning strategies are typically conducted in a centralized manner. However, this approach is often impractical, particularly for sensitive data like medical information, which is often distributed among clients and cannot be shared. This distributed scenario significantly complicates the fine-tuning process for CLIP.

Recent studies addressed this challenge by combining federated learning (FL) with fine-tuning techniques, a technique known as **federated fine-tuning**. Existing approaches (Xiao et al., 2023; Wang et al., 2024b; Marchisio et al., 2023; Chua et al., 2023; Khalid et al., 2023; Wang et al., 2024a) typically fine-tune these models without utilizing the entire model. Instead, they often employ layer-drop techniques (Sajjad et al., 2023) to compress a full model into a sub-model. The sub-model and an emulator are distributed to clients. Clients then update this compressed sub-model with their private data with the help of the emulator iteratively. The resulting sub-model is eventually incorporated into the full model to complete fine-tuning. However, these compression techniques fail to maintain alignment between the fine-tuned compressed layers and the remaining ones, leading to performance degradation. Federated parameter-efficient fine-tuning (PEFT) techniques, such as FedCLIP (Lu et al., 2023) and FedPETuning (Zhang et al., 2023), have emerged to address the aforementioned problem. These approaches involve deploying the foundation model with an additional adapter on each client, which is then collaboratively trained like FedAvg (McMahan et al., 2017). The aggregated adapter is subsequently integrated into the foundation model to achieve fine-tuning. However, they have several issues:

**Indistinguishable in domain-specific characteristics**. In real-world applications, the data collected by clients may exhibit different characteristics even for the same task. For instance, the stylistic realism of an image can vary across different forms of visual art, such as painting, photography, and digital art, leading to unique artistic expressions. However, existing models typically employ a single adapter

---

[1]The Pennsylvania State University [2]Sony AI. Correspondence to: Lingjuan Lyu <Lingjuan.lv@sony.com>, Fenglong Ma <fenglong@psu.edu>.

*Proceedings of the 42nd International Conference on Machine Learning*, Vancouver, Canada. PMLR 267, 2025. Copyright 2025 by the author(s).

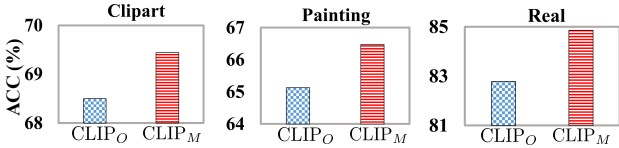

*Figure 1.* In-domain preliminary results.

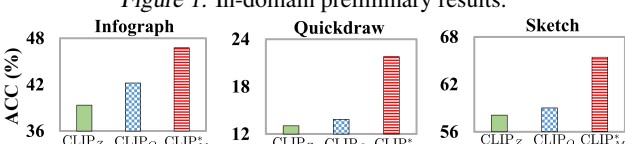

*Figure 2.* Out-of-domain preliminary results.

to capture knowledge from mixed domains, resulting in a performance gap compared to domain-specific adapters. We conducted a preliminary experiment on the DomainNet dataset to validate this observation, using three domains: "*clipart*", "*painting*", and "*real*". Following FedCLIP (Lu et al., 2023), we employed an attention-based adapter to fine-tune CLIP in two ways. The first way is to use all the data from three domains to fine-tune **one adapter**, denoted as $CLIP_O$. The other is to fine-tune **three individual adapters** only using each domain's data, denoted as $CLIP_M$.

The results are shown in Figure 1. It can be observed that despite using data from all three domains to fine-tune the adapter, $CLIP_O$ still performs worse than $CLIP_M$, which fine-tunes each adapter using only domain-specific data. This issue is expected to exacerbate in the federated fine-tuning setting due to the heterogeneity of clients, leading to an aggregated adapter inferior to centralized fine-tuning. These initial findings motivate us to develop domain-specific adapters for use in federated PEFT.

**Incapable to out-of-domain generalization**. While existing federated fine-tuning approaches can improve performance compared to zero-shot inference on the original models, they still struggle when faced with new or out-of-domain data. To explore the out-of-domain generalization ability, we use the **fine-tuned** $CLIP_O$ to directly conduct the inference on three new domains: "*infograph*", "*quickdraw*", and "*sketch*", comparing with the **zero-shot** inference with CLIP, denoted as $CLIP_Z$. We fine-tune **three new individual adapters** using each domain's data denoted as $CLIP_M^*$, which can be treated as the performance upper bound.

Figure 2 shows the preliminary results of the out-of-domain testing. We can observe that fine-tuning with a shared adapter ($CLIP_O$) does improve performance compared to $CLIP_Z$, but the degree of improvement is limited, as the results are far from the performance achieved by $CLIP_M^*$. Therefore, it is crucial to enhance the adapters' capability for out-of-domain generalization, especially in the federated fine-tuning setting.

However, addressing the aforementioned issue is challenging. On the one hand, it is hard to directly extend existing

work to model domain-specific characteristics. Sub-model fine-tuning approaches encounter difficulties in compressing multiple domain-specific sub-models and aggregating them. Similarly, PEFT approaches face challenges in aggregating adapters with diverse knowledge. On the other hand, equipping the capability of out-of-domain generalization with federated fine-tuning is an open challenge in this domain and is largely underexplored in existing studies. Thus, it is urgent to develop a new method to tackle these challenges simultaneously.

In this paper, we propose a novel federated fine-tuning approach named **Fed**erated **A**dapter **G**eneralization (FedAG), as shown in Figure 3. Intuitively, the types of domains for a specific task are usually limited, and the data belonging to a domain is usually collected by a specific client. This motivates us to design a new model under the **cross-silo** federated learning setting and allow all clients to be involved in the learning at each communication round. Besides, this setting also allows us to employ multiple fine-grained adapters to inject domain-specific knowledge into corresponding adapters while enhancing the capability of out-of-domain knowledge generalization by jointly combining these adapters. Unlike existing work, which either compresses a sub-model for each client or deploys a foundation model, we enable clients to have their domain-specific models representing the characteristics of their data. These client models are trained with private data (Figure 3(b)) and uploaded to the server to inject their domain-specific knowledge into CLIP.

## 2. Related Work

Fine-tuning foundation models is essential for specific downstream tasks, especially parameter-efficient fine-tuning (PEFT) (Ding et al., 2023; He et al., 2022; Han et al., 2024). However, the data privacy issue limits the fine-tuning of foundation models in a centralized way. Offsite-tuning (Xiao et al., 2023) incorporating federated learning techniques (McMahan et al., 2017) has recently been proposed to address this problem by incorporating federated learning techniques. Existing works can be roughly divided into three categories. Federated full model tuning (Deng et al., 2023; Fan et al., 2023) uses client outcomes as feedback to guide the fine-tuning of the foundation model. Federated partial model tuning (Peng et al., 2024; Marchisio et al., 2022; Khalid et al., 2023) compresses a submodel from the foundation model, then sends the compressed one to clients for extracting client knowledge, and finally aggregates the parameters learned from clients into the foundation model. Federated PEFT (Lu et al., 2023; Zhang et al., 2023) techniques use an extra adapter to learn and exchange client knowledge.

Our work falls into the federated PEFT category, and only a few studies have begun exploring this challenging task. In

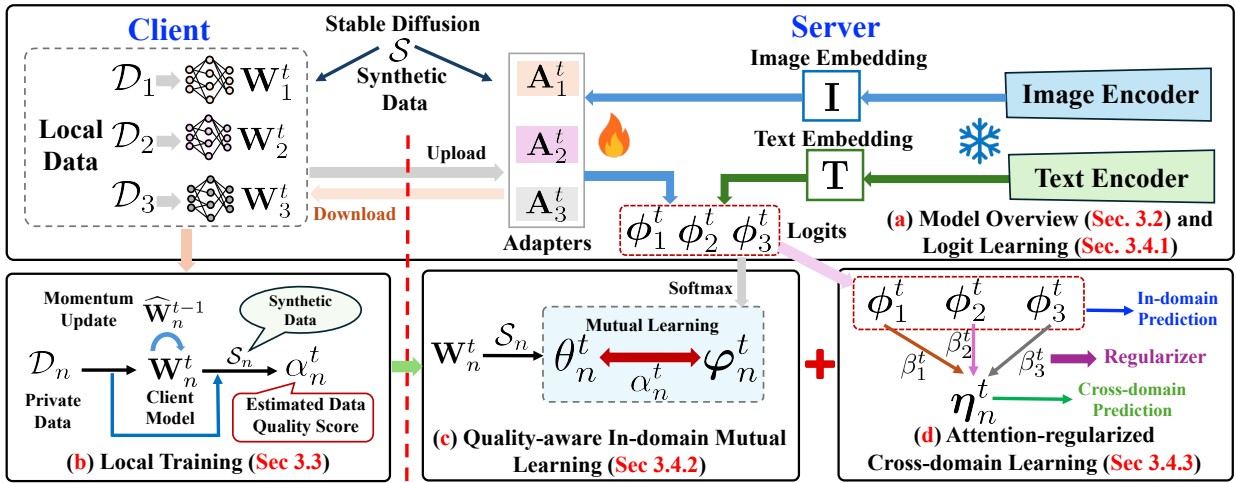

*Figure 3.* The overview of proposed FedAG. We use three domains ($N = 3$) as an example.

(Lu et al., 2023), each client has a foundation model and exchanges the adapters with the server in each communication round. The server conducts the basic FedAvg on the adapter and returns it to the clients. Similarly, FedPETuning (Zhang et al., 2023) provides a parameter-efficient tuning approach on pre-the trained language models via sharing part of the client models in federated learning. The aforementioned studies typically require clients to possess foundation models. However, this is impractical since small clients may be unable to fine-tune a large foundation model. Our approach places the foundation model on the server side, representing a more practical setting. Moreover, our objective is to enable clients to collaboratively contribute to the foundation model learning using their specific domain knowledge, without accessing local data.

## 3. Methodology

### 3.1. Model Input

The proposed model FedAG aims to iteratively inject domain knowledge into the vision foundation model deployed on the server through collaboration with $N$ *mutually exclusive and independent domain-specific* clients $\{C_1, \cdots, C_N\}$ without sharing their private data $\{\mathcal{D}_1, \cdots, \mathcal{D}_N\}$. To facilitate knowledge transfer while safeguarding clients' data privacy, the conventional approach involves data-free knowledge transfer, where often random Gaussian noise is utilized to distill knowledge from one model to another (Chen et al., 2019). Despite recent advancements (Raikwar & Mishra, 2022), noise-based knowledge transfer still encounters performance degradation compared to using real data. To conduct effective knowledge transfer, we leverage the open-source text-to-image model, Stable Diffusion 2.0 (Rombach et al., 2022), to generate domain-specific data $\mathcal{S}_n$ for each client $C_n$.

In practice, clients will share the style information (text prompt or the generated textual inversion token) so that domain-specific synthetic data $\{\mathcal{S}_1, \cdots, \mathcal{S}_N\}$ can be generated on the server[1]. Once synthetic data is generated, they will be transferred to the corresponding clients to perform the quality estimation. The communication of the synthetic data is only a one-time cost and is often negligible. The details of synthetic data generation can be found in §Sec. 4.1.

### 3.2. Model Overview

As shown in Figure 3, the proposed FedAG model comprises two main updates: the client update and the server update. The **client update** module (§Sec. 3.3) is designed to train a local model $f_n$ for each client $C_n$ using their respective data $\mathcal{D}_n$, where the parameters of $f_n$ (i.e., $\mathbf{W}_n^t$ at the $t$-th communication round) encapsulate the domain-specific knowledge. Additionally, it estimates a data-quality score $\alpha_n^{i,t} \in \boldsymbol{\alpha}_n^t$ for each synthetic instance $\mathbf{s}_n^i \in \mathcal{S}_n$. The client model parameters $\mathbf{W}_n^t$ and the estimated quality scores $\boldsymbol{\alpha}_n^t$ are then uploaded to the central server for further processing.

During the **server update** at the $t$-th communication round, FedAG first learns the logits of synthetic data using the CLIP framework in §Sec. 3.4.1. It then integrates the domain knowledge from $\mathbf{W}_n^t$ into the corresponding domain-specific attention-based adapter $\mathbf{A}_n^t$ based on the learned logits through a quality-aware *in-domain* mutual learning module that is detailed in §Sec. 3.4.2. Furthermore, it extends the model's capability to out-of-domain knowledge using an attention-regularized *cross-domain* learning module, as described in §Sec. 3.4.3. Afterward, the updated client

---

[1] Although it is possible to generate synthetic data on the client side and upload it to the server, some clients may lack the capacity to deploy the Stable Diffusion model. Furthermore, uploading data is less efficient than downloading it. Therefore, based on these considerations, we enable the server to generate synthetic data.

models (denoted as $\{\widehat{\mathbf{W}}_1^t, \cdots, \widehat{\mathbf{W}}_N^t\}$) are redistributed to their respective clients for another round of the client update.

## 3.3. Client Update

### 3.3.1. CLIENT MODEL TRAINING

At the $t$-th communication round, client $C_n$ will receive an updated model $\widehat{\mathbf{W}}_n^{t-1}$ from the server, which is trained using the synthetic data $\mathcal{S}_n$ in the server update. Since the generated synthetic data $\mathcal{S}_n$ is different from the real domain data $\mathcal{D}_n$, directly using $\widehat{\mathbf{W}}_n^{t-1}$ as the initialized client model at the $t$-the communication round (i.e., $\mathbf{W}_n^t = \widehat{\mathbf{W}}_n^{t-1}$) will be unsuitable[2]. To mitigate this issue, we propose the use of momentum update to initialize the client model as follows:

$$\mathbf{W}_n^t = \gamma \mathbf{W}_n^{t-1} + (1-\gamma)\widehat{\mathbf{W}}_n^{t-1}, \tag{1}$$

where $\gamma$ is the hyperparameter. We then use the traditional cross-entropy (CE) loss to train the client model's parameters $\mathbf{W}_n^t$ for the $n$-th client using $\mathcal{D}_n$ as follows:

$$\min_{\mathbf{W}_n^t} \mathcal{L}_n^t := \frac{1}{|\mathcal{D}_n|} \sum_{(\mathbf{x}_n^i, \mathbf{y}_n^i) \in \mathcal{D}_n} \mathrm{CE}(f_n(\mathbf{x}_n^i; \mathbf{W}_n^t), \mathbf{y}_n^i), \tag{2}$$

where $f_n$ is a ViT-Tiny model (Wu et al., 2022), $|\mathcal{D}_n|$ is the total number of private training data, $\mathbf{x}_n^i$ is the $i$-th data feature, $\mathbf{y}_n^i \in \{0,1\}^{|\mathcal{Y}|}$ is the corresponding label, and $\mathcal{Y}$ is the set of distinct labels, which is shared by all domains. The trained model $\mathbf{W}_n^t$ via Eq. (2) contains the knowledge of the $n$-th domain.

### 3.3.2. QUALITY ESTIMATION FOR DOMAIN-SPECIFIC SYNTHETIC DATA

The synthetic dataset $\mathcal{S}_n$, generated through stable diffusion, is essential for the server update but presents an *unknown* quality challenge. To address this, we propose estimating data quality using a prototype-based similarity measurement for each domain-specific set of generated data $\mathcal{S}_n$, utilizing the trained local model $\mathbf{W}_n^t$.

**Label-aware Prototype Representation Learning.** Let $\mathcal{D}_n^y \subset \mathcal{D}_n$ denote the subset of training data with labels $y \in \mathcal{Y}_n$. For each data instance $\mathbf{x}_n^i$ within $\mathcal{D}_n^y$, we first derive its feature representation $\mathbf{r}_n^{i,t}$ using the encoder layers of $\mathbf{W}_n^t$ before the prediction layer. We then compute a prototype representation $\mathbf{p}_n^{y,t}$ for each label category $y$ by averaging the representations of all data in $\mathcal{D}_n^y$, specifically, $\mathbf{p}_n^{y,t} = \frac{1}{|\mathcal{D}_n^y|} \sum_{\mathbf{x}_n^i \in \mathcal{D}_n^y} \mathbf{r}_n^{i,t}$.

**Similarity-based Data Quality Estimation.** For the generated data subset $\mathcal{S}_n^y \subset \mathcal{S}_n$ labeled $y$ in the $n$-th domain, each instance $\mathbf{s}_n^i \in \mathcal{S}_n^y$ also receives a feature representation

---

[2]An empirical experiment has been conducted to demonstrate this issue, as shown in **Appendix D**.

$\mathbf{q}_n^{i,t}$ through $\mathbf{W}_n^t$. We then calculate the cosine similarity $\alpha_n^{i,t}$ between $\mathbf{q}_n^{i,t}$ and the corresponding prototype $\mathbf{p}_n^{y,t}$, represented as $\alpha_n^{i,t} = \cos(\mathbf{q}_n^{i,t}, \mathbf{p}_n^{y,t})$. The vector of these similarity scores, $\boldsymbol{\alpha}_n^t$, for all generated data in $\mathcal{S}_n$ on the $n$-th client, is then compiled and prepared for uploading to the server along with $\mathbf{W}_n^t$.

This methodology offers significant advantages: it ensures that uploading synthetic data quality scores does not compromise the confidentiality of client data, and it allows each client model to provide specific data-quality scores, thus enhancing the precision of the mutual learning process (§Sec. 3.4.2).

## 3.4. Server Update

Upon receiving the uploaded client models $\{\mathbf{W}_1^t, \cdots, \mathbf{W}_N^t\}$ and their corresponding estimated data-quality scores $\{\boldsymbol{\alpha}_1^t, \cdots, \boldsymbol{\alpha}_N^t\}$, the server integrates domain-specific knowledge into CLIP. This is achieved by incorporating domain-specific attention-based adapters $\{\mathbf{A}_1^t, \cdots, \mathbf{A}_N^t\}$, each consisting of an identical multi-layer block positioned after the feature extractor of CLIP.

### 3.4.1. CLIP-BASED LOGIT LEARNING

The goal of FedAG is to inject domain knowledge included in client model parameters into the CLIP model in a PEFT way. Let $\mathrm{Enc}_{img}()$ represent the **frozen** image encoder and $\mathrm{Enc}_{txt}()$ be the **frozen** text encoder of CLIP. Let $\mathbf{L}_y$ denote the description of class label $y$, i.e., "a photo of a $[class]$". To learn the logit for an image $\mathbf{s}_n^i \in \mathcal{S}_n$, we follow the CLIP pre-training framework and take the image $\mathbf{s}_n^i$ and all the label descriptions $\{\mathbf{L}_y\}_{y=1}^{|\mathcal{Y}|}$ as the input. In particular, we first obtain the representations of $\mathbf{s}_n^i$ and $\mathbf{L}_y$ using the corresponding encoders as follows:

$$\mathbf{I}_n^i = \mathrm{Enc}_{img}(\mathbf{s}_n^i), \mathbf{T}_y = \mathrm{Enc}_{txt}(\mathbf{L}_y). \tag{3}$$

Following FedCLIP (Lu et al., 2023), the image representation $\mathbf{I}_n^i \in \mathbb{R}^d$ will pass an attention-based adapter $\mathbf{A}_n$ to obtain a fine-tuned domain-specific representation as follows:

$$\tilde{\mathbf{I}}_n^{i,t} = \mathbf{A}_n^t(\mathbf{I}_n^i) \odot \mathbf{I}_n^i = \mathrm{Softmax}(\mathrm{MLP}_n^{1,t}(\mathrm{Tanh}(\mathrm{MLP}_n^{2,t}(\mathbf{I}_n^i)))) \odot \mathbf{I}_n^i. \tag{4}$$

where $\tilde{\mathbf{I}}_n^{i,t} \in \mathbb{R}^d$, $d$ is the dimension size, and $\odot$ denotes the element-wise dot product. MLP is the multi-layer perception.

Finally, we can obtain the domain-specific logit for the input image as follows:

$$\phi_n^{i,t} = [\tilde{\mathbf{I}}_n^{i,t} \cdot \mathbf{T}_1^\top, \cdots, \tilde{\mathbf{I}}_n^{i,t} \cdot \mathbf{T}_{|\mathcal{Y}|}^\top]. \tag{5}$$

### 3.4.2. QUALITY-AWARE IN-DOMAIN MUTUAL LEARNING

To transfer domain-specific knowledge from the client model $\mathbf{W}_n^t$ to the CLIP model (i.e., the corresponding adapter $\mathbf{A}_n^t$), an intuitive way is to conduct knowledge distillation (Hinton et al., 2015) by treating $\mathbf{W}_n^t$ as the teacher network and the adapter-based CLIP as the student network. However, this simple strategy presents several limitations: it overlooks the quality of domain-specific synthetic data $\mathcal{S}_n$ involved in the distillation process and only allows uni-directional knowledge transfer, which does not update the local model $\mathbf{W}_n^t$, thus underutilizing the potential of the federated learning framework.

To overcome these shortcomings, we introduce a quality-aware in-domain mutual learning strategy. This approach not only ensures effective integration of domain-specific knowledge into $\mathbf{A}_n^t$ but also facilitates dynamic updates of the local model, leveraging the quality assessments of the synthetic data to enhance the overall learning process. Note that we use $\widehat{\mathbf{W}}_n^t$ to distinguish the updates of the client model $\mathbf{W}_n^t$ on the server. The loss function is defined as follows:

$$\min_{\mathbf{A}_n^t, \widehat{\mathbf{W}}_n^t} \mathcal{J}_n^t := \frac{1}{2\sum_{j=1}^{|\mathcal{S}_n|} \alpha_n^{j,t}} \sum_{\mathbf{s}_n^i \in \mathcal{S}_n} \alpha_n^{i,t} \Big\{ \mathrm{KL}(\boldsymbol{\theta}_n^{i,t} \| \boldsymbol{\varphi}_n^{i,t}) \quad (6)$$

$$+ \mathrm{KL}(\boldsymbol{\varphi}_n^{i,t} \| \boldsymbol{\theta}_n^{i,t}) \Big\}, \quad (7)$$

$$\boldsymbol{\theta}_n^{i,t} = f_n(\mathbf{s}_n^i; \widehat{\mathbf{W}}_n^t), \boldsymbol{\varphi}_n^{i,t} = \mathrm{softmax}(\boldsymbol{\phi}_n^{i,t})), \quad (8)$$

where $\boldsymbol{\theta}_n^{i,t}$ is the predicted probabilities by the client model $\widehat{\mathbf{W}}_n^t$ on each data instance $\mathbf{s}_n^i$ on the server, and $\boldsymbol{\varphi}_n^{i,t}$ is probabilities ouputed by the CLIP model using Eq. (5). $\mathrm{KL}(\cdot \| \cdot)$ is the Kullback–Leibler divergence.

### 3.4.3. ATTENTION-REGULARIZED CROSS-DOMAIN LEARNING

Using Eq. (7), we can update the adapters and client models simultaneously. However, such a design may only work for data belonging to existing domains, i.e., there is a lack of generalization ability for out-of-domain data. We propose a novel attention-regularized cross-domain learning strategy to equip the proposed FedAG with the capability for dealing with out-of-domain data.

In particular, for a synthetic data instance $\mathbf{s}_n^i \in \mathcal{S}_n$, we not only generate its logit $\boldsymbol{\phi}_n^{i,t}$ via Eq. (5) with the domain-specifc adaptor $\mathbf{A}_n^t$ but also from other adaptors $\{\mathbf{A}_1^t, \cdots, \mathbf{A}_{n-1}^t, \mathbf{A}_{n+1}^t, \cdots, \mathbf{A}_N^t\}$. We calculate the attention score $\beta_k^{i,t} \in \mathbb{R}$ ($k \in [1, N]$) for each adaptor using a softmax function on top of an MLP layer and then obtain

the aggregated logit for each data instance as follows:

$$\boldsymbol{\eta}_n^{i,t} = \sum_{k=1}^N \beta_k^{i,t} \boldsymbol{\phi}_k^{i,t}, \quad (9)$$

$$[\beta_1^{i,t}, \cdots, \beta_N^{i,t}] = \mathrm{softmax}([\mathrm{MLP}(\boldsymbol{\phi}_1^{i,t}), \cdots, \mathrm{MLP}(\boldsymbol{\phi}_N^{i,t})]). \quad (10)$$

The domain index $n$ is known for each training data during the training. Thus, the attention weight $\beta_n^{i,t}$ should be larger than those obtained from the other adapters. We use this intuition as prior knowledge to guide the model learning via an attention-based regularizer as follows:

$$\mathcal{R}_n^{i,t} = \max(0, \delta + \max([\beta_1^{i,t}, \cdots, \beta_{n-1}^{i,t}, \beta_{n+1}^{i,t}, \cdots, \beta_N^{i,t}]) - \beta_n^{i,t})), \quad (11)$$

where $\delta$ is the margin hyperparameter.

### 3.4.4. SERVER OPTIMIZATION

Based on Eqs. (7), (8), (9), and (11), we obtain the final loss function for the server update as follows:

$$\min_{\mathcal{A}^t, \mathcal{W}^t} \mathcal{G}^t := \frac{1}{N} \sum_{n=1}^N \Big[ \mathcal{J}_n^t + \sum_{(\mathbf{s}_n^i, \mathbf{y}_n^i) \in \mathcal{S}_n} \big[ \underbrace{\mathrm{CE}(\boldsymbol{\varphi}_n^{i,t}, \mathbf{y}_n^i)}_{\text{In-domain Prediction}}$$

$$+ \underbrace{\mathrm{CE}(\boldsymbol{\kappa}_n^{i,t}, \mathbf{y}_n^i)}_{\text{Cross-domain Prediction}} + \lambda \mathcal{R}_n^{i,t} \big] \Big], \quad (12)$$

where $\mathcal{A}^t = \{\mathbf{A}_1^t, \cdots, \mathbf{A}_N^t\}$, $\mathcal{W}^t = \{\widehat{\mathbf{W}}_1^t, \cdots, \widehat{\mathbf{W}}_N^t\}$, $\boldsymbol{\kappa}_n^{i,t} = \mathrm{softmax}(\boldsymbol{\eta}_n^{i,t})$, and $\lambda$ is the hyperparameter. The updated client models $\mathcal{W}^t = \{\widehat{\mathbf{W}}_1^t, \cdots, \widehat{\mathbf{W}}_N^t\}$ will be re-distributed to the corresponding domain-specific clients for update in the next communication round.

## 3.5. Inference

FedAG will be trained iteratively using Eqs. (2) and (12) until convergence. We then conduct the inference on the testing data. For the **in-domain** scenario, where the domain index $n$ is *known*, we use the label index with the maximum value in $\boldsymbol{\phi}_n^i$ as the predicted label, i.e., $\hat{y}_n^i = \arg\max_{\{1, \cdots, |\mathcal{Y}|\}}(\boldsymbol{\phi}_n^i)$ via Eq. (5). For the **out-of-domain** scenario where the domain is *unknown*, we use the label index with the maximum value in $\boldsymbol{\eta}^i$ as the predicted label, i.e., $\hat{y}^i = \arg\max_{\{1, \cdots, |\mathcal{Y}|\}}(\boldsymbol{\eta}^i)$ via Eq. (9).

## 4. Experiments

### 4.1. Experimental Setups

**Datasets.** To fairly validate the proposed model FedAG in our experiments, we focus on the image classification task on three commonly used domain-shifting datasets: Domain-Net, Office-Home, and ImageCLEF-DA. More details can be found in Appendix. We also incorporate synthetic data

Table 1. The results (mean and standard deviation) of in-domain evaluation of three runs.

| Setting | | Method | ImageCLEF-DA | | Office-Home | | | DomainNet | | |
|---|---|---|---|---|---|---|---|---|---|---|
| | | | Caltech | ImageNet | Art | Product | Real | Clipart | Painting | Real |
| Zero-shot | | $\text{CLIP}_Z$ | $97.25 \pm 1.03$ | $96.87 \pm 1.67$ | $78.12 \pm 1.21$ | $85.14 \pm 0.84$ | $86.33 \pm 1.25$ | $62.67 \pm 2.68$ | $59.77 \pm 2.50$ | $78.07 \pm 1.90$ |
| Centra. | Classical | $\text{ViT}_{cen}$ | $85.41 \pm 3.87$ | $82.06 \pm 2.89$ | $62.81 \pm 3.68$ | $83.97 \pm 3.91$ | $76.32 \pm 3.68$ | $53.44 \pm 3.97$ | $58.32 \pm 2.23$ | $77.01 \pm 2.65$ |
| | PEFT | $\text{CLIP}_L$ | $98.49 \pm 1.70$ | $95.45 \pm 2.44$ | $85.01 \pm 2.78$ | $87.92 \pm 2.06$ | $88.44 \pm 1.25$ | $68.15 \pm 1.89$ | $65.66 \pm 1.97$ | $83.28 \pm 2.05$ |
| | | $\text{CLIP}_A$ | $98.11 \pm 2.01$ | $95.52 \pm 1.42$ | $84.17 \pm 2.22$ | $88.02 \pm 1.76$ | $88.26 \pm 1.86$ | $68.5 \pm 2.20$ | $65.13 \pm 1.66$ | $82.79 \pm 2.37$ |
| Federated | Classical | FedAvg | $95.11 \pm 3.90$ | $83.33 \pm 2.74$ | $75.62 \pm 1.76$ | $86.85 \pm 2.58$ | $82.07 \pm 2.96$ | $51.66 \pm 3.80$ | $53.02 \pm 2.74$ | $69.34 \pm 3.97$ |
| | | $\text{FedAvg}_{ft}$ | $90.06 \pm 2.08$ | $80.25 \pm 2.30$ | $61.33 \pm 1.71$ | $75.51 \pm 2.09$ | $74.68 \pm 2.15$ | $48.27 \pm 3.60$ | $43.87 \pm 1.77$ | $62.06 \pm 2.08$ |
| | | FedProx | $95.75 \pm 3.62$ | $84.16 \pm 2.85$ | $76.98 \pm 1.52$ | $87.26 \pm 1.47$ | $83.15 \pm 1.70$ | $50.4 \pm 2.76$ | $53.45 \pm 1.05$ | $69.87 \pm 1.45$ |
| | | $\text{FedProx}_{ft}$ | $91.30 \pm 1.85$ | $80.54 \pm 2.11$ | $62.47 \pm 1.86$ | $75.65 \pm 1.38$ | $74.98 \pm 1.76$ | $48.89 \pm 3.10$ | $44.92 \pm 1.62$ | $63.77 \pm 1.10$ |
| | CLIP | FedCLIP | $97.34 \pm 3.11$ | $97.89 \pm 2.06$ | $82.14 \pm 1.35$ | $84.33 \pm 2.31$ | $87.62 \pm 1.89$ | $67.96 \pm 2.05$ | $65.78 \pm 1.66$ | $82.93 \pm 1.21$ |
| | | FedOT | $97.26 \pm 2.85$ | $97.91 \pm 1.73$ | $82.56 \pm 1.91$ | $85.47 \pm 2.98$ | $86.61 \pm 3.26$ | $67.68 \pm 2.74$ | $65.85 \pm 1.78$ | $83.20 \pm 1.40$ |
| | | FedAG | $\mathbf{98.62 \pm 1.34}$ | $\mathbf{98.56 \pm 1.78}$ | $\mathbf{84.97 \pm 1.78}$ | $\mathbf{88.69 \pm 1.06}$ | $\mathbf{88.79 \pm 1.57}$ | $\mathbf{70.36 \pm 1.96}$ | $\mathbf{66.29 \pm 1.08}$ | $\mathbf{84.92 \pm 0.85}$ |

during the model training. The number of synthetic data for each training domain equals 10% of the real domain data. The details of synthetic data generation for different datasets are as follows. When training the proposed FedAG, we also incorporate domain-level synthetic data generated by Stable Diffusion V2. The number of synthetic data for each training domain equals 10% of the real domain data. For the style-distinctive datasets, **DomainNet** and **OfficeHome**, synthetic data can be readily generated using text prompts following the template "a photograph/drawing of $class in $style style". However, for **ImageCLEF-DA**, where the style information is implicit and challenging to articulate using text prompts, we resort to generating synthetic data using textual inversion (Gal et al., 2022). Textual inversion entails deriving an appropriate text token corresponding to the implicit style. We sampled 10 instances from each of the 12 classes within the real ImageCLEF dataset and employed the Diffuser library to perform textual inversion. Once the style token is derived, the server utilizes a similar template, "a $class in $style_token style", to generate synthetic images for **ImageCLEF-DA**. Implementation Details can be found in **Appendix C**.

**Baselines.** We compare the proposed FedAG with several baselines in different settings, including zero-shot inference, centralized training, and federated learning.

- **Zero-Shot Inference.** We directly use the original CLIP model to predict the labels for given images in the testing data denoted as **$\text{CLIP}_Z$**.

- **Centralized Learning.** Since FedAG uses private domain data $\{\mathcal{D}_1, \cdots, \mathcal{D}_N\}$ for client training and synthetic data $\{\mathcal{S}_1, \cdots, \mathcal{S}_N\}$ for server training, for a fair comparison, we also use them together for the centralized training baselines. This setting involves two kinds of centralized training: classical centralized training and fine-tuning on CLIP. For the classical training, we directly train ViT with all data, denoted as **$\text{ViT}_{cen}$**. We also choose two commonly used parameter-efficient fine-tuning methods, adapter fine-tuning and LoRA (Hu et al., 2022) as baselines, which are denoted as **$\text{CLIP}_A$** and **$\text{CLIP}_L$**. $\text{CLIP}_A$

will learn a shared adapter, but the number of parameters in the adaptor is the same as that of FedAG, although FedAG is equipped with several domain-specific adapters. We set the rank for $\text{CLIP}_L$ as 32.

- **Federted Learning.** We use two classical federated learning approaches, **FedAvg** (McMahan et al., 2017) and **FedProx** (Li et al., 2020), as baselines. These approaches are trained only with client data without interacting with CLIP. Since our model FedAG uses synthetic data for fine-tuning the client models, in the experiments, we also fine-tuned FedAvg and FedProx the same epochs as our approach on the server at each communication round. The fine-tuned models are denoted as **$\text{FedAvg}_{ft}$** and **$\text{FedProx}_{ft}$**. The most relevant baselines are **FedCLIP** (Lu et al., 2023) and **FedOT** (Xiao et al., 2023). FedCLIP deploys a CLIP model for each client and fine-tunes the adapter on the local side. The adapters are uploaded to the server for aggregation, similar to FedAvg. To conduct a fair comparison, we conduct the same fine-tuning process on the adapter embedded in the same CLIP model on the server side as our own approach. FedOT (Xiao et al., 2023) is a federated version of Offsite-Tuning, where the CLIP model generates a compressed model and an emulator, which are shared with clients for their training.

### 4.2. In-domain Result Analysis

We train the models using the domains shown in the table and conduct the testing with the remaining domain data. Table 1 presents the **average accuracy of three runs** for the in-domain evaluation. We can observe that the proposed FedAG performs best on all domains in all datasets. $\text{CLIP}_Z$ is a zero-shot learning model with CLIP, which does not use any training data. It performs better than the classical federated learning models like FedAvg and FedProx. These comparisons prove the predictive power of foundation models for downstream tasks. The centralized PEFT approaches ($\text{CLIP}_L$ and $\text{CLIP}_A$) achieve comparable performance but outperform the zero-shot model $\text{CLIP}_Z$, which confirms the necessity of fine-tuning foundation models for boosting performance. Although they are trained in a centralized manner

Table 2. The results (mean and standard deviation) of out-of-domain evaluation of three runs.

| Setting | | Method | ImageCLEF-DA | Office-Home | DomainNet | | |
|---|---|---|---|---|---|---|---|
| | | | Pascal | Clipart | Infograph | Quickdraw | Sketch |
| Zero-shot | | $CLIP_Z$ | $82.13 \pm 0.58$ | $61.07 \pm 1.09$ | $39.34 \pm 2.24$ | $13.06 \pm 1.87$ | $58.11 \pm 1.56$ |
| Centra. | Classical | $ViT_{cen}$ | $71.66 \pm 3.40$ | $42.66 \pm 2.74$ | $20.15 \pm 3.66$ | $10.67 \pm 1.62$ | $40.75 \pm 2.93$ |
| | PEFT | $CLIP_L$ | $81.22 \pm 1.63$ | $67.15 \pm 2.88$ | $42.10 \pm 2.87$ | $14.38 \pm 2.70$ | $59.48 \pm 2.60$ |
| | | $CLIP_A$ | $81.08 \pm 2.97$ | $67.31 \pm 2.51$ | $42.22 \pm 3.51$ | $13.85 \pm 1.55$ | $59.01 \pm 2.74$ |
| Federated | Classical | FedAvg | $78.33 \pm 4.33$ | $43.58 \pm 3.65$ | $26.75 \pm 3.70$ | $10.78 \pm 2.95$ | $40.56 \pm 3.11$ |
| | | $FedAvg_{ft}$ | $73.02 \pm 3.88$ | $41.12 \pm 3.20$ | $24.27 \pm 3.14$ | $10.33 \pm 2.40$ | $37.91 \pm 2.25$ |
| | | FedProx | $78.69 \pm 3.06$ | $45.88 \pm 2.87$ | $27.50 \pm 3.55$ | $12.04 \pm 2.29$ | $40.97 \pm 2.84$ |
| | | $FedProx_{ft}$ | $72.68 \pm 2.74$ | $40.75 \pm 2.44$ | $24.63 \pm 2.81$ | $11.89 \pm 2.48$ | $38.54 \pm 2.36$ |
| | CLIP | FedCLIP | $82.45 \pm 2.46$ | $64.44 \pm 2.53$ | $41.65 \pm 3.18$ | $12.89 \pm 2.70$ | $59.23 \pm 2.52$ |
| | | FedOT | $82.10 \pm 2.89$ | $65.27 \pm 2.69$ | $40.70 \pm 2.90$ | $15.51 \pm 2.64$ | $60.30 \pm 2.70$ |
| | | FedAG | $\mathbf{83.78 \pm 1.94}$ | $\mathbf{68.15 \pm 1.08}$ | $\mathbf{45.56 \pm 2.26}$ | $\mathbf{21.04 \pm 2.20}$ | $\mathbf{63.29 \pm 1.76}$ |

and perform the best among all baselines, their performance is worse than that of FedAG. The reason is that these two approaches only use one adapter or two low-rank matrices to store mixed domain knowledge. However, our model uses domain-specific adapters to capture the characteristics of domains, thus leading to the best performance in the in-domain evaluation. These results also validate the design of multiple domain adapters. When comparing with the federated fine-tuning approaches, we can find they also perform better than $CLIP_Z$ but have performance gaps with centralized PEFT approaches $CLIP_L$ and $CLIP_A$. These results demonstrate the efficacy of injecting domain knowledge into foundation models in a federated way.

### 4.3. Out-of-domain Result Analysis

The ultimate goal of training a foundation model is to apply it to various downstream tasks, including inference on unseen data. To assess this capability, we conduct an out-of-domain evaluation using the trained models with in-domain evaluation to validate the unseen domains. The results are shown in Table 2.

We observe similar trends to those in the in-domain evaluation. Specifically, FedAG outperforms all baselines, and $CLIP_Z$ performs better than classical models. However, compared to the in-domain evaluation results, the performance gaps between the centralized PEFT models (i.e., $CLIP_L$ and $CLIP_A$) and $CLIP_Z$ are not as significant. In fact, their performance is even worse than that of FedOT in several domains. These results highlight the limitations of existing models in generalizing out-of-domain knowledge. In contrast to existing approaches, our proposed FedAG consistently achieves superior performance, leading to significant improvements in accuracy. For instance, in the Quickdraw domain of the DomainNet dataset, our approach demonstrates a 36% performance increase compared to the

Table 3. Ablation study on the DomainNet dataset. C for Clipart, P for Painting, R for Real, I for Infograph, Q for Quickdraw, S for Sketch.

| Method | In-domain | | | Out-of-domain | | |
|---|---|---|---|---|---|---|
| | C | P | R | I | Q | S |
| $FedAG_{mome}$ | 68.54 | 65.60 | 83.00 | 44.38 | 20.14 | 62.85 |
| $FedAG_{quality}$ | 68.12 | 65.13 | 83.11 | 44.79 | 20.58 | 63.15 |
| $FedAG_{kd}$ | 69.88 | 64.27 | 82.55 | 42.62 | 18.40 | 62.03 |
| $FedAG_{cross}$ | 70.04 | 66.11 | 84.13 | 40.63 | 15.70 | 59.04 |
| $FedAG_{reg}$ | 68.26 | 64.05 | 81.08 | 42.01 | 17.55 | 61.69 |
| FedAG | **70.36** | **66.29** | **84.92** | **45.56** | **21.04** | **63.29** |

best baseline FedOT. These results strongly indicate that our model effectively handles out-of-domain knowledge.

### 4.4. Abaltion Study

We use the following baselines to validate the effectiveness of our model design. $FedAG_{mome}$ does not use momentum update (i.e., Eq. (1)) for the local model after receiving the learned global model. $FedAG_{quality}$ denotes removing data quality estimation in Eq. (7). $FedAG_{cross}$ denotes removing the module of attention-regularized cross-domain learning. $FedAG_{reg}$ means that we remove the designed attention-based regularization term $\mathcal{R}$ in Eq. (12). The results of the ablation studies are presented in Tables 3. It is evident that removing each designed module results in a performance drop, underscoring the necessity of each module. Interestingly, the in-domain results suggest that cross-domain learning may not be as crucial compared to momentum updates and data quality estimation. However, in the out-of-domain evaluation, $FedAG_{cross}$ plays a significant role, as its removal leads to a dramatic performance drop. These findings align with the motivations behind our model design, emphasizing the importance of the cross-domain learning module in addressing the out-of-domain issue.

## 4.5. Momentum Update for Clients

Figure 7 displays the empirical experiment results of models trained with real and synthetic data on the DomainNet dataset in a centralized manner using three domains: "Clipart", "Painting", and "Real", where the model used in this preliminary experiment is the same as our client model, which is TinyViT (Wu et al., 2022). The testing data used in the experiment are the head-out in-domain data. The data details can be found in §Sec. 4.1.

It is evident from Figure 7 that models trained with real data outperform those trained with synthetic data by a significant margin. Therefore, replacing the well-trained client model $\mathbf{W}_n^{t-1}$ with the distributed $\widetilde{\mathbf{W}}_n^{t-1}$ arbitrarily would disrupt the clients' training. Thus, we propose to use the momentum update for the client training.

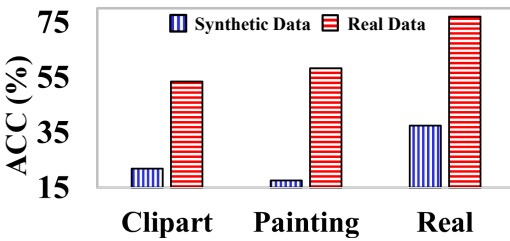

*Figure 4.* Performance comparison with synthetic data and real data.

## 4.6. Influence of Domain Knowledge

To investigate the impact of domain knowledge on in-domain and out-of-domain performance, we adjust the setting of in-domain and out-of-domain data utilization. We selected DomainNet for our experiments because it encompasses a wide variety of domains, and its zero-shot performance across these domains exhibits significant variability. In our primary experiment, we assigned **three** domains Clipart, Painting, and Real to different clients for training. We further extended our study to include three additional training configurations: (1) **two** training domains: Clipart and Painting, (2) **four** training domains: Clipart, Painting, Real and Infograph, and (3) **five** training domains: Clipart, Painting, Real, Infograph, and Quickdraw. For all the scenarios, we always keep the *Sketch* domain for out-of-domain testing. The in-domain and out-of-domain test results from these configurations are subsequently reported. From the analysis presented in Table 4, it is evident that increasing the number of domains enhances the performance in both in-domain and out-of-domain tests. This improvement is particularly noticeable in domains that initially demonstrated weaker performance, such as Infograph and Quickdraw. Such enhancements can be attributed to our proposed method's ability to effectively utilize knowledge from various domains.

*Table 4.* Results with different training domains.

| Setting | Clipart | Painting | Real | Infograph | Quickdraw | Sketch |
|---|---|---|---|---|---|---|
| $2 \to 4$ | 69.23 ✓ | 64.33 ✓ | 81.15 ✗ | 42.90 ✗ | 16.20 ✗ | 61.00 ✗ |
| $3 \to 3$ | 70.36 ✓ | 66.29 ✓ | 84.92 ✓ | 45.56 ✗ | 21.04 ✗ | 63.29 ✗ |
| $4 \to 2$ | 70.21 ✓ | 66.57 ✓ | 85.17 ✓ | 45.48 ✓ | 21.89 ✗ | 65.08 ✗ |
| $5 \to 1$ | 70.35 ✓ | 66.48 ✓ | 85.68 ✓ | 46.50 ✓ | 23.84 ✓ | 66.78 ✗ |

## 4.7. Generalization Study

To assess the effectiveness of various image encoders and to validate the generalizability of our proposed method, we conducted additional tests using different CLIP image encoders on the server side, including ResNet-50 (**RS**) and ViT-Tiny (**VT**). The outcomes of these tests are shown in Figure 5. In our main experiments, we employed the ViT-B-32 (**VB**) encoder. We can observe that ResNet-50 is the weakest encoder, but its performance is comparable to that of the best baseline FedOT listed in Tables 1 and 2. Using other encoders can significantly enhance the performance, validating the generalizability of `FedAG`.

## 4.8. Performance Upper-bound Exploration

To further validate the efficacy of our proposed method, `FedAG`, we conducted a comparative analysis against an upper bound benchmark. In the upper bound scenario, all training and synthetic data are combined to fine-tune a CLIP adapter, with each adapter tailored to a specific domain. This process, referred to as $\text{CLIP}_M$, involves tuning domain-specific adapters using the corresponding domain data. The comparative results are presented in Table 9. Our findings reveal that the performance of `FedAG` closely approaches, and in some instances surpasses, this upper bound. This further demonstrates the effectiveness of our proposed `FedAG`.

*Table 5.* Upper bound analysis.

| **Dataset** | **Domain** | $\text{CLIP}_M$ | `FedAG` |
|---|---|---|---|
| ImageCLEF-DA | Caltech | 98.55 | 98.62 |
| | ImageNet | 95.86 | 98.56 |
| | Pascal | 83.67 | 83.78 |
| Office-Home | Art | 85.14 | 84.97 |
| | Product | 88.78 | 88.69 |
| | Real | 88.98 | 88.79 |
| | Clipart | 69.59 | 68.15 |
| DomainNet | Clipart | 69.44 | 70.36 |
| | Painting | 66.47 | 66.29 |
| | Real | 84.86 | 84.92 |
| | Infograph | 46.78 | 45.56 |
| | Quickdraw | 21.80 | 21.04 |
| | Sketch | 65.44 | 63.29 |

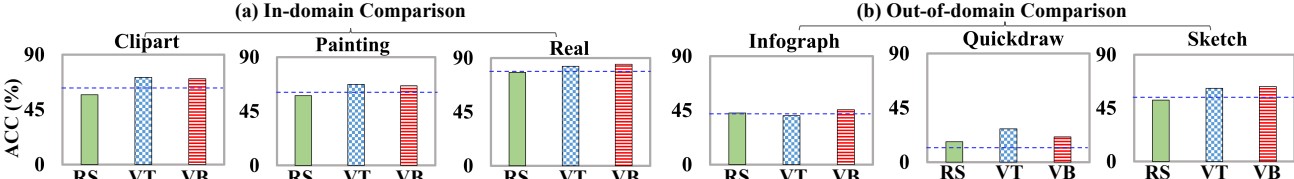

*Figure 5.* Results on the DomainNet dataset with different image encoders of CLIP. The blue dashed line denotes the performance of the best baseline FedOT using ViT-B-32 (**VB**) as the encoder.

*Table 6.* The performance of `FedAG` with different sizes of synthetic data. "in" means the in-domain results, and "out" means the out-of-domain results.

| Dataset | Domain | Data Volume | | | |
|---|---|---|---|---|---|
| | | **100%** | **75%** | **50%** | **25%** |
| Image CLEF-DA | Caltech (in) | 98.62 | 98.88 | 97.51 | 97.86 |
| | ImageNet (in) | 98.56 | 98.47 | 97.22 | 97.31 |
| | Painting (out) | 83.78 | 83.04 | 82.50 | 82.43 |
| Office -Home | Art (in) | 84.97 | 84.30 | 82.46 | 82.04 |
| | Product (in) | 88.69 | 88.21 | 87.51 | 86.77 |
| | Real (in) | 88.79 | 88.63 | 87.18 | 86.45 |
| | Clipart (out) | 68.15 | 67.30 | 66.78 | 66.44 |
| DomainNet | Clipart (in) | 70.36 | 68.41 | 67.96 | 66.12 |
| | Painting (in) | 66.29 | 65.15 | 64.04 | 61.50 |
| | Real (in) | 84.92 | 84.50 | 83.09 | 81.96 |
| | Infograph (out) | 45.56 | 44.21 | 43.76 | 40.05 |
| | Quickdraw (out) | 21.04 | 20.76 | 18.89 | 16.76 |
| | Sketch (out) | 63.29 | 62.33 | 61.07 | 59.53 |

### 4.9. Synthetic Data Volume

In this subsection, we examine the influence of synthetic data volume on the performance of our proposed algorithm. We sampled subsets of 75%, 50%, and 25% from the synthetic data used in our main experiments while keeping all other settings constant. The results for both in-domain and out-of-domain evaluations are presented in Table 10. From the analysis, we observe the following: (1) As the volume of synthetic data decreases, there is a corresponding decline in performance across the three datasets for both in-domain and out-of-domain scenarios. (2) The performance degradation from reducing synthetic data from 50% to 25% is more pronounced than the drop from 100% to 50%. (3) Notably, even with a minimal amount of synthetic data (25%), our approach maintains reasonable performance in both in-domain and out-of-domain settings for all datasets. In summary, our investigation into the effects of synthetic data volume confirms its impact on algorithm performance; however, our approach demonstrates resilience to reduced data volumes within certain limits.

### 4.10. Case Study

To further illustrate the effectiveness of the proposed `FedAG` for out-of-domain generalization, we use two case studies to

visualize the attention weights $\beta$ learned by Eq. (9) with the four fine-tuned models on various domains, as mentioned in Table 4. These variations are depicted in Figure 6. The first image, originating from the Sketch domain, shows that within the context of out-of-domain data, Clipart receives the highest attention score, followed by Real and Painting. Conversely, for the in-domain example from the Real domain (as depicted in the second image), including identical domain data in the training regimen results in it receiving the highest attention score. These observations, supported by both Table 4 and Figure 6, effectively illustrate the dynamic nature of domain knowledge in influencing model performance. They also highlight how the attention score $\beta$ adjusts in response to various domain knowledge configurations, emphasizing the adaptability and efficiency of our method in leveraging diverse domain insights.

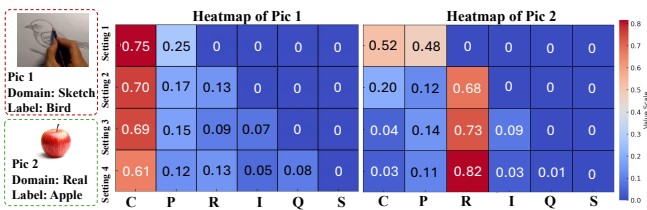

*Figure 6.* The visulization of $\beta$ on two image samples.

## 5. Conclusion

In this study, we introduced `FedAG`, an innovative federated fine-tuning approach designed to address the challenges of domain-specific characteristics and out-of-domain generalization with CLIP. Using multiple fine-grained adapters and novel learning modules, `FedAG` effectively integrates domain-specific knowledge and enhances generalization across diverse domains. Our extensive experiments on various datasets validate the efficacy of `FedAG`, showing performance improvements over state-of-the-art methods. Our proposed approach has the potential to be applied to cross-device federated learning if a large number of domains are available, which we plan to extend in our future work.

## Impact Statement

This paper contributes to advancing federated learning, foundation models, and domain adaptation by enabling efficient collaboration among multiple parties. Our work facilitates the development of foundation model generalization while addressing challenges in distributed data utilization, domain adaptation, and parameter-efficient fine-tuning. These contributions have broad implications for sustainable computing, improving the adaptability and efficiency of foundation models across diverse and decentralized environments.

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

# Appendix

## A. Pseudo-code of `FedAG`

Algorithm 1 shows the pseudo-code of the proposed `FedAG` model, which contains two main updates: the client update (lines 6-14) and the server update (lines 15 - 32).

## B. Dataset

**DomainNet**[3]. It totally has 569,010 images from 6 domains, including clipart, infographics, painting, quickdraw, real, and sketch. Each domain contains 48K to 172K images, categorized into 345 classes.

**Office-Home**[4]. It has 15,500 images from 4 different dimensions: artistic images, clip art, product images, and real-world images. Each domain has 65 object classes.

**ImageCLEF-DA**[5]. It is a benchmark for the ImageCLEF 2014 domain adaption challenge, including Caltech-256, ImageNet ILSVRC 2012, and Pascal VOC 2012. There are 12 categories and 50 images in each category.

## C. Implementation Details

For each dataset, we assign each in-domain data to one client. We utilize ViT-Tiny-patch16-224[6] for the client model and ViT-B-32[7] for the image encoder for the server side. Our experimental setup involves 10 communication rounds. For the local update, we set the local training epoch as 10, the local learning rate as 1e-4, the batch size as 32, $\gamma = 0.9$, and the optimizer used in the optimization as Adam. For the server update, we set $\lambda = 0.1$, $\delta$ = 1e-3, the epoch of quality-aware in-domain mutual learning as 3, and the epoch of adapter initialization as 5. We keep the shared parameter settings consistent across our method and the baselines. The unique hyperparameters for the baselines are adopted as per the specifications in their respective original papers. All experiments are conducted on an NVIDIA A6000 with CUDA version 12.0, running on a Ubuntu 20.04.6 LTS server. All baselines and the proposed `FedAG` are implemented using PyTorch 2.0.1.

## D. Momentum Update for Clients

Figure 7 displays the empirical experiment results of models trained with real and synthetic data on the DomainNet dataset in a centralized manner using three domains: "Clipart", "Painting", and "Real", where the model used in this preliminary experiment is the same as our client model, which is TinyViT (Wu et al., 2022). The testing data used in the experiment are the head-out in-domain data. The data details can be found in §Sec. 4.1.

It is evident from Figure 7 that models trained with real data outperform those trained with synthetic data by a significant margin. Therefore, replacing the well-trained client model $\mathbf{W}_n^{t-1}$ with the distributed $\widehat{\mathbf{W}}_n^{t-1}$ arbitrarily would disrupt the clients' train-

---

[3] https://ai.bu.edu/M3SDA/
[4] https://www.hemanthdv.org/officeHomeDataset.html
[5] https://www.imageclef.org/2014
[6] https://huggingface.co/timm/vit_base_patch16_224.augreg2_in21k_ft_in1k
[7] https://huggingface.co/openai/clip-vit-base-patch32

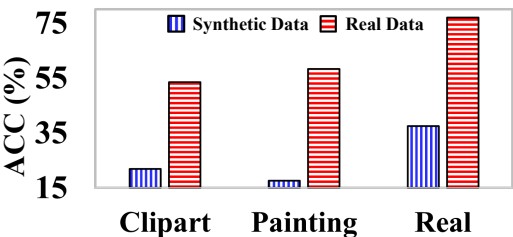

*Figure 7.* Performance comparison with synthetic data and real data.

ing. Thus, we propose to use the momentum update for the client training.

## E. Details of Baselines

Table 7 shows the approaches used in the experiments, including training ways, deployment details, and used data.

## F. Performance Upper-bound Exploration

To further validate the efficacy of our proposed method, `FedAG`, we conducted a comparative analysis against an upper bound benchmark. In the upper bound scenario, all training and synthetic data are combined to fine-tune a CLIP adapter, with each adapter tailored to a specific domain. This process, referred to as $\text{CLIP}_M$, involves tuning domain-specific adapters using the corresponding domain data. The comparative results are presented in Table 9. Our findings reveal that the performance of `FedAG` closely approaches, and in some instances surpasses, this upper bound. This further demonstrates the effectiveness of our proposed `FedAG`.

## G. Additional Experiment Results of Ablation Study

In order to more thoroughly evaluate the effectiveness of our designed modules, we conducted ablation studies on two additional datasets: ImageCLEF-DA and Office-Home. The results of these studies are detailed in Table 8. Our observations indicate that the performance trends on ImageCLEF-DA and Office-Home datasets are consistent with those observed on the DomainNet dataset. This consistency across different datasets further validates the effectiveness of the individual components of our proposed method, `FedAG`, reinforcing the robustness and generalizability of our approach.

## H. Study on Key Hyperparameters

We examined the impact of key hyperparameters $\lambda$ and $\delta$ on our model's performance, with results illustrated in Figure 8 for $\lambda$ and Figure 9 for $\delta$. Our observations from the experiments are as follows: (1) Increasing values of $\lambda$ and $\delta$ correspond to improved in-domain performance. This enhancement can be attributed to the mechanisms described in equations 11 and 12, where higher values of $\lambda$ and $\delta$ focus the training more intensively on in-domain knowledge acquisition. (2) Within a certain range, elevating $\lambda$ and $\delta$ also enhances out-of-domain learning, potentially due to the effectiveness of in-domain adapters, which facilitate out-of-domain inference through the cross-domain learning module discussed in Sec. 3.4.3. However, excessively high values of $\lambda$ or $\delta$ might overly concentrate the learning process on in-domain aspects, thereby

*Table 7.* Summaries of approaches used in this work.

| Name | Training Way | Deployment Details | Used Data |
|---|---|---|---|
| $CLIP_Z$ | No training | Zero-short on CLIP | No data |
| $CLIP_O$ | Centralized | Fine-tuning one adapter on CLIP | In-domain data |
| $CLIP_M$ | Centralized | Fine-tuning multiple individual adapters | In-domain data |
| $CLIP_M^*$ | Centralized | Fine-tuning multiple individual adapters | Out-of-domain data |
| $CLIP_L$ | Centralized | Fine-tuning one adapter on CLIP | In-domain and synthetic data |
| $CLIP_A$ | Centralized | Fine-tuning LoRA on CLIP | In-domain and synthetic data |
| $ViT_{cen}$ | Centralized | Fine-tuning ViT-B-32 | In-domain and synthetic data |
| FedAvg | Federated | Client: Fine-tuning ViT-Tiny | In-domain data |
| | | Server: No training | No data |
| FedProx | Federated | Client: Fine-tuning ViT-Tiny | In-domain data |
| | | Server: No training | No data |
| $FedAvg_{ft}$ | Federated | Client: Fine-tuning ViT-Tiny | In-domain data |
| | | Server: Fine-tuning the aggregated model | Synthetic data |
| $FedProx_{ft}$ | Federated | Client: Fine-tuning ViT-Tiny | In-domain data |
| | | Server: Fine-tuning the aggregated model | Synthetic data |
| FedCLIP | Federated | Client: CLIP + Adapter | In-domain data |
| | | Server: No training | No data |
| | | Post-training: Fine-tuning one adapter | Synthetic data |
| FedOT | Federated | Client: Adapter + emulator | In-domain data |
| | | Server: Fine-tuning one adapter on CLIP | Synthetic data |
| FedAG | Federated | Client: Fine-tuning ViT-Tiny | In-domain data |
| | | Server: Fine-tuning multiple adapters on CLIP | Synthetic data |

*Table 8.* Ablation study experiment results on ImageCLEF-DA and Office-Home datasets. ✓ denotes the in-domain and ✗ denotes the out-of-domain.

| Method | ImageCLEF-DA | | | Office-Home | | | |
|---|---|---|---|---|---|---|---|
| | Caltech ✓ | ImageNet ✓ | Pascal ✗ | Art ✓ | Product ✓ | Real ✓ | Clipart ✗ |
| $FedAG_{mome}$ | 98.28 | 98.17 | 83.11 | 84.88 | 88.09 | 88.20 | 66.15 |
| $FedAG_{quality}$ | 97.45 | 98.21 | 83.26 | 82.47 | 87.14 | 87.24 | 67.20 |
| $FedAG_{kd}$ | 97.41 | 98.04 | 83.21 | 81.89 | 86.54 | 86.57 | 67.31 |
| $FedAG_{cross}$ | 98.44 | 98.36 | 82.55 | 83.09 | 87.88 | 88.62 | 63.89 |
| $FedAG_{reg}$ | 97.30 | 97.13 | 81.17 | 81.55 | 85.69 | 86.11 | 67.36 |
| FedAG | **98.62** | **98.56** | **83.78** | **84.97** | **88.69** | **88.79** | **68.15** |

degrading out-of-domain performance. In summary, careful adjustment of $\lambda$ and $\delta$ offers a means to tailor performance across different domain configurations, achieving a balance between in-domain focus and out-of-domain generalization.

## I. Synthetic Data Volume

In this subsection, we examine the influence of synthetic data volume on the performance of our proposed algorithm. We sampled subsets of 75%, 50%, and 25% from the synthetic data used in our main experiments while keeping all other settings constant. The results for both in-domain and out-of-domain evaluations are presented in Table 10. From the analysis, we observe the following: (1) As the volume of synthetic data decreases, there is a corresponding decline in performance across the three datasets for both in-domain and out-of-domain scenarios. (2) The performance degradation from reducing synthetic data from 50% to 25% is

more pronounced than the drop from 100% to 50%. (3) Notably, even with a minimal amount of synthetic data (25%), our approach maintains reasonable performance in both in-domain and out-of-domain settings for all datasets. In summary, our investigation into the effects of synthetic data volume confirms its impact on algorithm performance; however, our approach demonstrates resilience to reduced data volumes within certain limits.

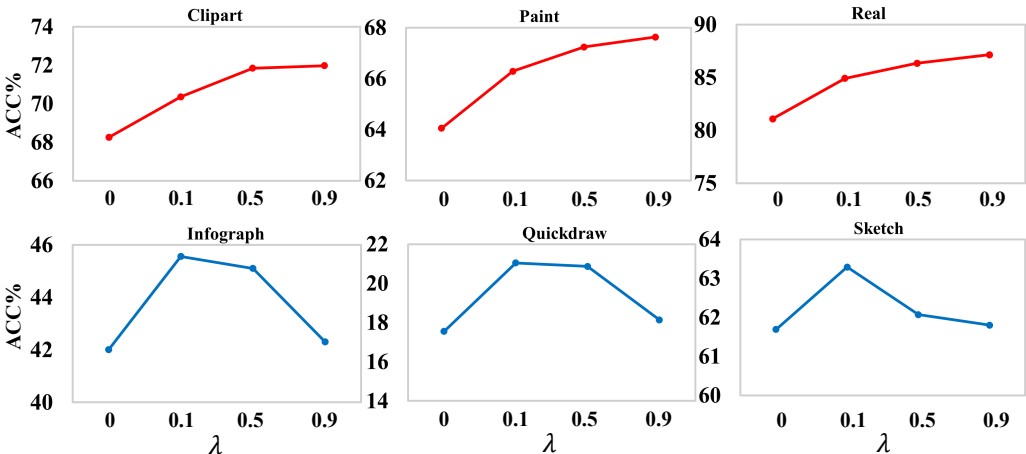

Figure 8. Hyperparameter study of $\lambda$ on the DomainNet dataset. Red is the color for the in-domain, and blue is the color for the out-of-domain.

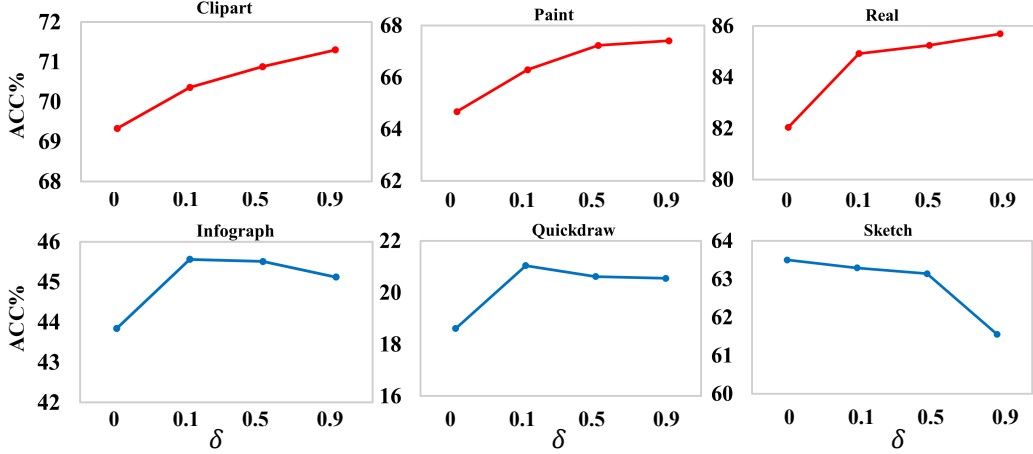

Figure 9. Hyperparameter study of $\delta$ on the DomainNet dataset. Red is the color for the in-domain, and blue is the color for the out-of-domain.

Table 9. Upper bound analysis.

| Dataset | Domain | $\text{CLIP}_M$ | FedAG |
|---|---|---|---|
| ImageCLEF-DA | Caltech | 98.55 | 98.62 |
| | ImageNet | 95.86 | 98.56 |
| | Pascal | 83.67 | 83.78 |
| Office-Home | Art | 85.14 | 84.97 |
| | Product | 88.78 | 88.69 |
| | Real | 88.98 | 88.79 |
| | Clipart | 69.59 | 68.15 |
| DomainNet | Clipart | 69.44 | 70.36 |
| | Painting | 66.47 | 66.29 |
| | Real | 84.86 | 84.92 |
| | Infograph | 46.78 | 45.56 |
| | Quickdraw | 21.80 | 21.04 |
| | Skectch | 65.44 | 63.29 |

Table 10. The performance of FedAG with different sizes of synthetic data. "in" means the in-domain results, and "out" means the out-of-domain results.

| Dataset | Domain | Data Volume | | | |
|---|---|---|---|---|---|
| | | 100% | 75% | 50% | 25% |
| Image CLEF-DA | Caltech (in) | 98.62 | 98.88 | 97.51 | 97.86 |
| | ImageNet (in) | 98.56 | 98.47 | 97.22 | 97.31 |
| | Painting (out) | 83.78 | 83.04 | 82.50 | 82.43 |
| Office -Home | Art (in) | 84.97 | 84.30 | 82.46 | 82.04 |
| | Product (in) | 88.69 | 88.21 | 87.51 | 86.77 |
| | Real (in) | 88.79 | 88.63 | 87.18 | 86.45 |
| | Clipart (out) | 68.15 | 67.30 | 66.78 | 66.44 |
| DomainNet | Clipart (in) | 70.36 | 68.41 | 67.96 | 66.12 |
| | Painting (in) | 66.29 | 65.15 | 64.04 | 61.50 |
| | Real (in) | 84.92 | 84.50 | 83.09 | 81.96 |
| | Infograph (out) | 45.56 | 44.21 | 43.76 | 40.05 |
| | Quickdraw (out) | 21.04 | 20.76 | 18.89 | 16.76 |
| | Sketch (out) | 63.29 | 62.33 | 61.07 | 59.53 |

---

**Algorithm 1** Algorithm Flow of $\mathtt{FedAG}$.

---

**Input:** Local data $\{\mathcal{D}_1, \cdots, \mathcal{D}_N\}$, Stable Diffusion V2, domain descriptions, task label descriptions, communication rounds $T$, local training epoch $E_c$, server training epoch $E_s$, and hyperparameters: $\lambda$, $\gamma$, and $\delta$

**Server Initialization**

    Use Stable Diffusion V2 to generate domain-specific synthetic data $\{\mathcal{S}_1, \cdots, \mathcal{S}_N\}$ based on domain descriptions and task label descriptions;

    Use CLIP training to initialize domain-specific adapters using $\{\mathcal{S}_1, \cdots, \mathcal{S}_N\}$ individually;

Distribute each domain-specific synthetic dataset $\mathcal{S}_n$ to the corresponding client $C_n$;

**for** each communication round $t = 1, 2, \cdots, \text{T}$ **do**

    **Client Update**

        **for** each client $n \in [1, \cdots, N]$ **do**

            Momentum update $\mathbf{W}_n^t$ if $t > 1$;

            **for** each local epoch $e \in [1, E_c]$ **do**

                | Update $\mathbf{W}_n^t$ with Eq. (2);

            **end**

            Calculate quality scores $\boldsymbol{\alpha}_n^t$ using $\mathbf{W}_n^t$ for $\mathcal{S}_n$;

        **end**

    Upload $\{(\mathbf{W}_1^t, \boldsymbol{\alpha}_1^t), \cdots, (\mathbf{W}_N^t, \boldsymbol{\alpha}_N^t)\}$ to the server;

    **Server Update**

        $\widehat{\mathbf{W}}_1^t = \mathbf{W}_1^t, \cdots, \widehat{\mathbf{W}}_N^t = \mathbf{W}_N^t$;

        **for** each server epoch $e \in [1, E_s]$ **do**

            Initialize the total loss $\mathcal{G}^t = 0$;

            **for** each domain $n \in [1, \cdots, N]$ **do**

                // Quality-aware In-domain Mutual Learning

                Obtain domain-specific logits $\boldsymbol{\phi}_n^{i,t}$ for each synthetic data $\mathbf{s}_n^i \in \mathcal{S}_n$ using Eq. (5);

                Obtain predicted propobalities $\boldsymbol{\theta}_n^{i,t}$ for each synthetic data $\mathbf{s}_n^i \in \mathcal{S}_n$ using the upload client model $\widehat{\mathbf{W}}_n^t$;

                Calculate $\mathcal{J}_n^t$ using $\boldsymbol{\theta}_n^{i,t}$ and $\boldsymbol{\phi}_n^{i,t}$ using Eq. (7) for each data;

                // Attention-regularized Cross-domain Learning

                Caluate the aggregated logits $\boldsymbol{\eta}_n^{i,t}$ using Eq. (9) each synthetic data $\mathbf{s}_n^i \in \mathcal{S}_n$;

                Obtain the attention-based regularize $\mathcal{R}_n^{i,t}$ using Eq. (11) each synthetic data $\mathbf{s}_n^i \in \mathcal{S}_n$;

                Obtain the loss of a specific domain $\mathcal{G}_n^t$;

                $\mathcal{G}^t += \mathcal{G}_n^t$;

            **end**

            Optimize $\mathcal{G}^t$ using Eq. (12);

        **end**

        Distribute the updated local models $\mathcal{W}^t = \{\widehat{\mathbf{W}}_1^t, \cdots, \widehat{\mathbf{W}}_N^t\}$ to the corresponding local clients;

**end**

---

