# OpenReview forum: "Enhancing Foundation Models with Federated Domain Knowledge Infusion"
_ICML.cc/2025/Conference — ICML 2025 poster_

### Official Review · Reviewer_373f · 2025-03-07

**Overall Recommendation:** 3

**Summary:**

This paper proposes an efficient federated fine-tuning approach that enhances out-of-domain generalization. In the proposed framework, each client utilizes a lightweight ViT model, which is trained on local data. Subsequently, data quality scores are computed using synthetic data and transmitted to the server along with the locally trained model. The server then performs mutual learning to distill knowledge from a large model (CLIP) and applies attention-regularized cross-domain learning to improve out-of-domain generalization.

**Claims And Evidence:**

By leveraging synthetic data generated by a diffusion model, the proposed method effectively distills knowledge from the large model to the client model. Experimental results validate the effectiveness of this approach.

**Essential References Not Discussed:**

Some relevant related works are missing. Efficient fine-tuning in FL has already been explored in federated prompt tuning (e.g., [ICML'24], [CVPR'24]). A comparison with these methods should be included in the paper.

[CVPR'24]] Unlocking the Potential of Prompt-Tuning in Bridging Generalized and Personalized Federated Learning

[ICML'24] Harmonizing Generalization and Personalization in Federated Prompt Learning

**Experimental Designs Or Analyses:**

It would be beneficial if the authors conduct additional experiments with various combinations of training domains while comparing the proposed method against other baselines.

**Methods And Evaluation Criteria:**

The approach of adopting a small model on each client while utilizing a large model on the server is reasonable. Regarding evaluation criteria, this paper employs standard benchmark datasets (e.g., DomainNet) to demonstrate the effectiveness of the proposed method.

**Other Comments Or Suggestions:**

The cross-silo FL setting targeted in this paper may have sufficient computing resources, as it generally assumes participation from large institutions such as hospitals. Therefore, wouldn't it be more appropriate to target the cross-device FL setting instead?

**Other Strengths And Weaknesses:**

### Strengths
- The approach of utilizing a small model on each client while leveraging a large model on the server is practically important.
- The idea of enhancing the local model by distilling knowledge from the large model on the server is promising.

### Weaknesses
- Some parts of the description of the proposed method are unclear. An additional MLP layer is required to compute $\beta$ in Eq. (10). Are these layers optimized during training?
- If the label set differs across clients, this label information should be shared with the server. However, this may lead to the leakage of each client's private information.
- One of the key challenges in FL is addressing data heterogeneity. However, this paper does not take this aspect into account. It would be beneficial to include experiments that consider data heterogeneity (e.g., using a Dirichlet distribution).
- It would be beneficial to clearly differentiate the unique contributions or advantages of this method from existing federated prompt learning approaches (e.g., [ICML'24], [CVPR'24]).
- To demonstrate the robustness of the proposed method across various scenarios, it would be beneficial to compare the results from different training domain combinations in Table 4 with existing baselines.
- The inference process of this method assumes that the model has to know whether a given test task belongs to an in-domain or out-of-domain scenario, which is impractical in real-world applications. During test time, the model does not have access to this information.

[CVPR'24]] Unlocking the Potential of Prompt-Tuning in Bridging Generalized and Personalized Federated Learning

[ICML'24] Harmonizing Generalization and Personalization in Federated Prompt Learning

**Questions For Authors:**

- See the sections above.

**Relation To Broader Scientific Literature:**

The unique contribution of this paper lies in utilizing a small model for each client and enhancing its performance by distilling knowledge from the large model on the server.

**Theoretical Claims:**

There is no theoretical claim in this paper.

---

> ### Author Rebuttal · Authors · 2025-03-25
>
> We thank the comments and questions.
>
> `>>> W1`
>
> Yes, the MLP layers are optimized during the training. We will clarify it in the final version.
>
> `>>> W2`
>
> We do not share any label information with the server.  As introduced in Sec 3.1, clients only share the style information via the text prompt (style is clear) or the generated textual inversion token ( style is vague), for the data generation.
>
> `>>> W3`
>
> We clarify that the domain shift among clients also represents the data heterogeneity in FL [1,2]. In our experiments, the data from each client represents a unique domain and there is no crossover/mixup between clients. Under such a heterogeneous setting, our approach is still able to help enhance the capability of the foundation models with the help of local clients’ domain knowledge. Furthermore, we add more experiments of the data class-heterogeneity. With ImageCLEF-DA maintaining settings unchanged in Table 1 and Table 2,  we partition the dataset with respect to the classes using the Dirichlet distribution in [3] to simulate the class heterogeneity.  The performance of the results decreases slightly but is still higher than the classical methods.
>
> |        | Caltech | ImageNet | Pascal |
> |--------|---------|----------|--------|
> | FedAG  | 97.61   | 97.15    | 82.66  |
>
> [1] Heterogeneous federated learning: State-of-the-art and research challenges. 2023.
>
> [2] A review of federated learning methods in heterogeneous scenarios. 2024.
>
> [3] Harmonizing Generalization and Personalization in Federated Prompt Learning, ICML 2024
>
> `>>> W4`
>
> [CVPR’24] designs a prompt-tuning approach to solve the data heterogeneity challenge in FL. They learn shared prompts at the server side and match them with the local groups and assign group prompts to them. The main design covers how to learn shared prompts, group prompts, prompt selection, and an optimization method to iteratively learn different knowledge. In [ICML’24]  finds the balance between generalization and personalization for the data heterogeneity. The approach is mainly a prompt-based approach, which is based on utilizing local personalized prompts with the help of a global prompt and an adaption term. Also, most of their designed modules are at the local side and they focus on the evaluation on the local clients.
>
> Compared with them, we have 2 key differences: 1, motivation part: we focus more on how to enhance the capability of foundation models at the server side with the help from the local clients by knowledge infusion; 2, method part: our key method is not a prompt-based approach and most operations happen at the server side to release the computation burden at the client side.
>
> `>>> W5`
>
> The results in Table 1 and 2 are already cover the setting of 3--->3 in Table 4. Due to the limited time during the rebuttal period to run experiments, we report the 2--->4 setting as below. (We omit the decimal part due to limited space)
>
> |        | Clipart | Painting | Real  | Info | Quick | Sketch |
> |-------------|--------|---------|------|-------|--------|--------|
> | FedAvg      | 48  | 48   | 66 | 22 | 10 | 35  |
> | FedAvg_ft   | 46  | 47   | 66 | 21 |  9 | 33  |
> | FedProx     | 48  | 50   | 67 | 23| 10 | 35  |
> | FedProx_ft  | 46  | 49   | 66 | 22 |  9  | 34  |
> | FedCLIP     | 65  | 61   | 72 | 37 | 11  | 46  |
> | FedOT       | 64 | 61   | 73 | 36 | 12  | 47  |
> | FedAG       | 69  | 64   | 81 | 42 | 16  | 61  |
>
> We see (1) given 2 in-domain combinations for training, the overall performance decreases compared with the setting where we have 3. This is because of the less training data, which causes the performance degradation; (2) Our approach still outperforms other baselines.
>
> `>>> W6`
>
> Our method can also work even without knowing the in-domain and out-of-domain of a given task and it can be scaled with more domains. Given the initialized number of adapters equal to the number of domains, our designed modules are capable of handling out-of-domain tasks. In particular, we report the results of different training and testing domains in Table 4, where we scale our approach to different numbers of domains. As the reviewer mentioned, given the assumption if we may not be able to access to the domain information that it is in-domain or out-of-domain and minimize the change of current approach, we could equally treat the data and use the label index with the maximum value in $\eta^{i}$ as the predicted label via eq 9. We provide the results in this case following the setting in Table 4 as below.
>
> |         | Clipart | Painting | Real  |
> |---------|---------|----------|-------|
> | Known   | 70.36   | 66.29    | 84.92 |
> | Unknown | 68.31   | 65.77    | 84.07 |
>
> The performance has a slight decrease in a certain range but still outperforms the baselines. The results show the effectiveness of our method given the condition without knowing the domain information during the inference process.
>
> We hope our reply sufficiently answers your question and addresses your concern.

---

> > ### Comment · Reviewer_373f · 2025-04-02
> >
> > I appreciate the authors for their efforts in responding to my concerns. However, I still have remaining concerns regarding W2, W3, W6.
> >
> > 1) W2: My original question was whether the label information should be shared with the server when the label sets differ across clients. For example, given 10 classes ranging from 0 to 9, one client may have samples from classes 0 to 3, while another may have classes 3 to 5. In such cases, should the server be aware of each client's label distribution? If it does, it can lead to leakage of the client information.
> >
> > 2) W3: Thank you for providing additional experiments under heterogeneous settings. However, it remains unclear whether the proposed method consistently outperforms the baselines in this setting.
> >
> > 3) W6: Although the authors provide promising results for the case where domain information is not given, I believe that the main results in the paper should also include those of the proposed method without access to domain information. This is important to support the claim that the paper addresses out-of-distribution generalization, where the target domain is assumed to be unknown.
> >
> > Overall, considering these concerns, I maintain my original score.

---

> > > ### Author Response · Authors · 2025-04-04
> > >
> > > We sincerely appreciate the reviewer's time in reading and responding to our rebuttal. We are pleased to have addressed some of your concerns. Due to the character limit in our previous response, we would like to take this opportunity to address the remaining points as follows:
> > >
> > > `>>> W2`
> > >
> > > Thanks for the question. Our proposed model does not require clients to share any label information with the server, even when the clients have different label sets. When generating synthetic data, the server generates the data for all labels, thus no client-specific label information is needed. Also, the server does not require client-specific label information to conduct the server update.
> > >
> > >  In our original setting, we focus on the domain shift problem and assume each client holds data with all labels. The setting you mentioned is the label heterogeneity issue, where each client may have different labels. We would like to further explain the client operations under this setting. The server  distributes a synthetic dataset $S_n$ to each client, which contains the generated data with all labels $ {y_1, …, y_Y}$ . Assume that the client n only has data with label $ y_3$  and $ y_5$ , we can only obtain the prototype representation $ p_3$  and $ p_5$  (line 212, Page 4). However, the prototypes of other classes are unknown. To address this issue, we use the average representation from all the client data as the shared prototype of the remaining classes. In this way, we can estimate the similarity score for each synthetic data (with all labels) and do not need to share the specific label distribution information with the server. The results of this setting can be seen in the response of W3.
> > >
> > > We appreciate your question and will add the details to the final version of our paper. We hope our response sufficiently addresses your concern.
> > >
> > > `>>> W3`
> > >
> > > Thank you for your comments. As per your suggestion, we provided more comparisons with baselines below. Note that we keep all settings consistent with the previous experiments described in our paper.
> > >
> > >
> > > | Method      | Caltech | ImageNet | Pascal |
> > > |-------------|---------|----------|--------|
> > > | FedAvg      | 88.65   | 78.07    | 72.54  |
> > > | FedAvg_ft   | 85.19   | 75.34    | 67.08  |
> > > | FedProx     | 89.34   | 79.65    | 73.88  |
> > > | FedProx_ft  | 85.78   | 78.28    | 73.10  |
> > > | FedCLIP     | 95.00   | 94.05    | 80.62  |
> > > | FedOT       | 95.97   | 93.66    | 81.91  |
> > > | **FedAG**   | **97.61** | **97.15**  | **82.66** |
> > >
> > > We can observe that the proposed FedAG outperforms baselines under the heterogeneous setting. However, we can also observe performance drops compared with the setting used in the original paper. The basic models, such as FedAvg and FedPro,x are sensitive to data heterogeneity, which leads to worse performance. Compared with FedCLIP and FedOT, our proposed model can leverage the generated data and the weighted regularization attention mechanism to capture the common knowledge and adaptively learn the diverse information across different clients under the data heterogeneity setting.
> > >
> > > We will add the results and related analysis in the final version of our paper. We hope our response adequately resolves your concern.
> > >
> > > `>>> W6`
> > >
> > > We greatly appreciate the reviewer’s acknowledgment of the added experimental results. As suggested, we will incorporate this section into the final version of the main paper. Specifically, we plan to:
> > >
> > > - In Sec 3.5, we will explain how to conduct inference without knowing the domain information as we introduced in our last rebuttal reply;
> > >
> > > - In Sec 4, we will introduce a new subsection (4.9) to explore the scenario where domain information is unavailable. We will include the additional experiments presented in our rebuttal and the related discussion to highlight the out-of-domain generalization capabilities of our proposed approach.
> > >
> > > We are grateful for the reviewer’s valuable suggestion and sincerely hope that our response answers the concern. Besides, we respectfully hope you to reconsider your Overall Recommendation of our submission.

---

### Official Review · Reviewer_otAj · 2025-03-12

**Overall Recommendation:** 3

**Summary:**

This paper introduces FedAG, a federated learning method to enhance vision foundation models (e.g., CLIP) by fine-tuning them across distributed domains while preserving data privacy. FedAG employs multiple domain-specific adapters, synthetic data generation via Stable Diffusion, and quality-aware mutual learning to capture domain knowledge. It also uses attention regularization to improve out-of-domain generalization. Experiments on ImageCLEF-DA, Office-Home, and DomainNet show FedAG outperforms centralized and federated baselines, achieving higher accuracy in both in-domain and out-of-domain settings.

---
I appreciate the authors for their rebuttal and will keep my rating.

**Claims And Evidence:**

Yes.

**Essential References Not Discussed:**

No.

**Experimental Designs Or Analyses:**

Yes.

**Methods And Evaluation Criteria:**

Yes.

**Other Comments Or Suggestions:**

The mutual learning and attention regularization modules (Sec. 3.4.2–3.4.3) are overly technical; intuitive diagrams or simplified explanations would improve accessibility.

**Other Strengths And Weaknesses:**

### Strengths:
1. The integration of multiple domain-specific adapters within the federated framework is interesting and seems novel.
2. The proposed method has practical value since it focuses on cross-silo federated learning and aligns with real-world privacy constraints.
3. The idea of using a diffusion model for synthetic data generation for privacy-preserving is interesting.

### Weaknesses:
1. The paper assumes a fixed number of domains, how to deal with dynamic or numerous clients? The computational overhead of managing multiple adapters is also unexplored.
2. The paper assumes the synthetic data is high-quality and representative, which lacks in-depth analysis. Poorly generated data could bias adapters or harm generalization.

**Questions For Authors:**

1. How does FedAG handle scenarios with a large or dynamic number of domains (e.g., 100+ clients)? Does the server-side adapter aggregation scale efficiently?
2. How does the quality of Stable Diffusion-generated data impact performance? Are there safeguards against adversarial or biased synthetic samples?

**Relation To Broader Scientific Literature:**

This paper is related to vision foundation models and federated learning.

**Theoretical Claims:**

Yes.

---

> ### Author Rebuttal · Authors · 2025-03-31
>
> We genuinely appreciate the reviewer’s valuable comments and questions. We would like to address them as follows for your review.
>
> `>>> W1` and `>>> Q1`
>
> Thanks for your constructive comment and question. If we have a dynamic or numerous clients, we will group clients by the domains they belong to. In Sec 3.4.2 or Figure 3.c, we can conduct a basic model aggregation approach to get the aggregated $\bar{W}_n$ for each domain group n, e.g., average-based approach. Then we can replace the original $W_n^t$ with the domain group-level \$\bar{W}_n$ The aggregation process will merge the knowledge from the same domain and can be generally plugged into the current proposed framework. Also, as it is a basic average operation of the model parameters, it may not raise too much extra computational burden. In our proposed approach, the number of the adapters is equal to the number of domains, and each domain adapter can only interact with one aggregated model parameter, as we discussed above.  However, we totally agree with the reviewer’s comments about the setting with a large number of domains. Our current work is based on the cross-silo setting and we have stated that we will further explore the potentials of extending this approach to the cross-device setting in the conclusion part (line 434-439). We hope our reply can sufficiently answer your question and address your concern.
>
> `>>> W2` and `>>> Q2`
>
> Thank you for the comment. We have actually considered the issues of synthetic data quality problems and discussed them in Sec. 3.3.2. We totally agree that the quality of generated data would have an effect on the performance. To solve this, in our proposed work, we design an estimation mechanism to estimate the quality of the generated data in Sec 3.3.2. In particular, we first obtain the prototype representation for each label category. After that, we calculate the cosine similarity of the representation of the generated data and the prototype to obtain the score \alpha. With this estimation, we add it as a weight to equations (6) and (7) (lines 245 to 248) to conduct mutual learning using quality assessments of the generated data. With a low quality of the synthetic data, the weight will be small and help control its effect on the overall performance.
>
>
> Furthermore, to examine the effectiveness of this quality evaluation module, we also provide the ablation study in Sec 4.4, and the results are shown in Table 3. We remove the quality estimation module for the synthetic data and name the setting as FedAG_quality. We observe that the performance drops in the evaluation of the in-domain and out-of-domain data, which demonstrates the effectiveness of the quality evaluation module to help control the utilization of the synthetic data.
>
>
> We do appreciate that the reviewer mentioned the possibility of adversarial samples in the synthetic data. We will explore this direction to further improve the security and bias of our proposed approach in future work. We hope our response fully clarifies your question and resolves your concern.

---

### Official Review · Reviewer_agi1 · 2025-03-13

**Overall Recommendation:** 5

**Summary:**

This paper introduces a federated learning approach to enhance the capability of foundation models to handle in-domain and out-of-domain tasks. In particular, the authors designed quality-aware in-domain mutual learning and attention-based cross-domain learning to capture the knowledge effectively. In this paper, they provided extensive experiment results and compared them with other baselines.

**Claims And Evidence:**

The claims in this paper are well supported by statements, formulation, experiments, and discussion.

**Essential References Not Discussed:**

No

**Experimental Designs Or Analyses:**

The authors validate the method via appropriate experimental design and results. In Table 1 and 2, they reported the in-domain and out-of-domain results along with other baselines and settings. They also conduct ablation study, generalization study, and case study. The visualization in Figure 5 facilities the understanding of out-of-domain generalization with the proposed approach.

**Methods And Evaluation Criteria:**

The designed method fits the problem setting. First of all, federated learning is able to solve the challenge of distributed data and the foundation model deployment. Secondly, the design of the adapter cluster considers the in-domain and out-of-domain respectively.

**Other Comments Or Suggestions:**

No

**Other Strengths And Weaknesses:**

Strengths
1, This paper studies a more practical setting to enhance the capability of foundation models via federated learning. It does not require the clients to be equipped with large models.
2, The in-domain and out-of-domain learning methods consider the different features of the knowledge from the data. The design is able to capture the information effectively.
3, This approach is straightforward and easy-to-implement. Based on the experiments and design, it can be well generalized to other scenarios and applications.
4, The whole pipeline only trains very limited parameters, which consider the efficiency as well.

Weaknesses
1, What amount of the synthetic data is used and how do they affect the results?
2, The paper lacks a pseudo algorithm for the whole pipeline.

**Questions For Authors:**

1, Could you please clarify how the stable diffusion generates the synthetic data in this work?

**Relation To Broader Scientific Literature:**

This paper explores a more practical setting where the foundation models are put at the server side and the local models interact with the large models via the adapter cluster. This setting can be extended to broader applications and scenarios.

**Theoretical Claims:**

The methodology part is formulated and described clearly.

---

> ### Author Rebuttal · Authors · 2025-03-31
>
> We genuinely appreciate the reviewer’s valuable comments and suggestions. We would like to address them as follows for your review.
>
> `>>> W1`
>
> In our main experiment, the amount of the synthetic data is equal to 10% of the real data for each domain. To further examine how the amount of synthetic data affects the results, we conduct a further study about the synthetic data volume in Appendix I. In particular, we randomly sample 75%, 50%, and 25% from the existing synthetic data in our main experiment and repeat the experiments while keeping all the settings unchanged. We observe that the performance will degrade with the reduction of the synthetic data. With very limited synthetic data (25%), our approach is still able to maintain an acceptable performance. The results demonstrate how the amount of the synthetic data affects the final results.
>
>  `>>> W2`
>
> Due to the limited page number in the main paper, we put the pseudo algorithm in the last page of the Appendix. To further enhance the readability of our work, we will try to put a simplified version of the pseudo code in the main paper.
>
>  `>>> Q`
>
> In our proposed framework, the clients share very limited and vague information with the server. We let the clients provide the style information via the text prompt or the generated textual inversion token. In particular, for easily distinguishable styles such as “Ghibili cartoon”, “Pablo Picasso”, direct text prompt can be used. For vague or ambiguous styles, one can use textual inversion, a technique that enables Stable Diffusion to learn a new embedding vector to represent this style from just a few sample images. This involves a brief training process, the system optimizes only this new embedding vector. It adjusts the vector so that when it's fed into the frozen Stable Diffusion model along with descriptive prompts (like "a photo in <my-new-style>"), the model generates images that look like the sample images. After the style tokens are gathered from clients, the server can simply apply the template such as “a clock in <style-token> style” as text prompt input to Stable Diffusion to generate synthetic data.
>
> Besides that, to avoid the effects of the low-quality synthetic data, we further design a quality estimation module to measure the quality of the generated data and add that score to our optimization equation. We will add a more detailed description to clarify this part in the final version of our paper.

---

> > ### Comment · Reviewer_agi1 · 2025-04-05
> >
> > The authors have addressed my concerns, I will raise my score.

---

### Official Review · Reviewer_9HNy · 2025-03-18

**Overall Recommendation:** 4

**Summary:**

This manuscript addresses the challenge of fine-tuning large-scale vision-language models in a federated learning setting under domain shifts. The authors propose FedAG (Federated Adapter Generalization), a method that introduces multiple domain-specific adapters to capture heterogeneous domain knowledge while maintaining out-of-domain generalization working under a federated setting.

The FedAG took a two step approach A) Quality-Aware In-Domain Mutual Learning step which utilizes client models’ knowledge to refine the adapter for each domain, weighting synthetic data by estimated quality; B) Attention-Regularized Cross-Domain Learning step which synthesize logits from all adapters for inference on out-of-domain inputs, guided by an attention-based regularizer to help identify which adapter’s knowledge is most relevant to new data.

Experimental evaluations on DomainNet, Office-Home, and ImageCLEF-DA demonstrate improvements in both in-domain accuracy and out-of-domain generalization over existing baselines such as FedCLIP, FedOT, and classical parameter-efficient fine-tuning approaches.

**Claims And Evidence:**

The authors claim that the proposed method will be an important step for generalising CLIP especially in handling out-of-domain predictions.

The method was evaluated on DomainNet, Office-Home, and ImageCLEF-DA to demonstrate improvements in both in-domain accuracy and out-of-domain generalization over existing baselines such as FedCLIP, FedOT, and classical parameter-efficient fine-tuning approaches.

**Essential References Not Discussed:**

Not I am aware of.

**Experimental Designs Or Analyses:**

The experimental design was overall sound and the data analyses was accurate.

**Methods And Evaluation Criteria:**

The evaluation method and criteria were proper.

**Other Comments Or Suggestions:**

The figure 3 was not very easy to understand. I assume the purpose of the chart is to give readers an intuitive overview before diving into the details in the text. I would suggest to keep it high-level to explain the steps in the training cycle.

There are some typos in the manuscript, e.g. in pseudo code of the algorithm "Caluate the aggregated logits...".

**Other Strengths And Weaknesses:**

The manuscript identifies the drawbacks of using a single adapter for data aggregated from multiple (and potentially very diverse) domains. Multiple domain-specific adapters align well with the stated aim to capture each domain’s particularities while still leveraging each other for out-of-domain generalization.

The authors present experiments on three domain-adaptation benchmarks (DomainNet, Office-Home, ImageCLEF-DA). They provide ablation studies (momentum, quality estimation, cross-domain learning, attention regularization) and hyperparameter sensitivity analyses, giving the reader a detailed view of how each component influences final performance.

**Questions For Authors:**

1. while the FedAG will maintain domain-specific adapters for each client, what will be additional computation and communication cost in comparison to the other methods?
2. In the real-world scenario, the domain shift might be more subtle than the difference in experimental dataset. When will it still be advantageous to take the proposed approach? Are there scenarios that the proposed approach may perform worth than prior approaches, e.g. PEFT?
3. The performance of Zero-shot inference was stronger than some classical approaches in both in-domain and out-of-domain inference, is there any explanation on this observation?

**Relation To Broader Scientific Literature:**

This paper presents a novel federated fine-tuning approach for large pretrained models, specifically CLIP. Unlike traditional methods like FedAvg (McMahan et al., 2017) and FedProx (Li et al., 2020) which aggregate fully trained local models, or Offsite-Tuning (Xiao et al., 2023) and FedCLIP (Lu et al., 2023) which distribute compressed or partial models, this work proposes a system where the server hosts CLIP and multiple adapters, while clients have a lighter ViT-Tiny. Clients collect domain-specific knowledge and transfer it to the server's adapters through a structured process, contrasting with prior single-adapter or sub-model compression strategies.

**Theoretical Claims:**

The manuscript did not take any theoretical claims. It was an empirical research.

---

> ### Author Rebuttal · Authors · 2025-03-31
>
> We sincerely appreciate the reviewer’s constructive suggestions and comments. We would like to reply respectively as follows.
>
> `>>> Response to Other Comments Or Suggestions`
>
> Thank you for the feedback. We will follow your suggestion to improve the figure and fix the typos in the final version of the paper.
>
> `>>> Q1`
>
> Thank you for the question. In our proposed approach, the number of adapters is equal to the number of domains. If we have more clients, we can group them into different groups based on the domains to which they belong. Then we aggregate the model parameters into one for each domain before we conduct the mutual learning process in our design. As for the computational cost, we freeze the encoders of the foundation models and only allow the parameters of adapters and the local model to be trainable and updated. Furthermore, we put most of our designed operations on the server side to release the burden of the local clients, as it is more common that the server side may have more flexible computation constraints than the local side.  As for the communication cost, as we mentioned in Sec 3.1, we may need to transfer the synthetic data from the server to the clients in a one-time manner, which can be negligible.  We appreciate the reviewer’s question and will investigate how to further reduce the computation and communication cost in future work. We hope our response can sufficiently address your concern.
>
> `>>> Q2`
>
> Thank you for the comment. It is possible that the real-world scenarios could be more subtle, but we believe that our method can work for these different scenarios as demonstrated with three datasets having different levels of domain shifts. In our approach, we proposed a cross-domain learning mechanism to capture the relationship among different domains and quantify it with a weight score $\beta$. Besides that, we add a regularizer to further adjust the attention based on the domains of the data.   As the reviewer commented, given a subtle domain shift in the real-world scenario, PEFT could be a more effective way. However,  we may think our method is still advantageous because we do not directly access the local data, while centralized PEFT does not have such advantages. We will explore this direction with some real world data and compare it with PEFT in our future work. We hope our reply can sufficiently answer your question.
>
> `>>> Q3`
>
> Thanks for the observation and the valuable question.  We would like to provide our explanations as follows. First of all, as described in Appendix C, we use the ViT-B-32 for the image encoder on the server side. As it is pretrained, it has the basic zero-shot inference capability for image-related tasks. Secondly, for the baseline ViT_cen, we put the data from the domains together and tune all the parameters, which shows the degraded performance. One possible reason could be that we fine-tune all the parameters of this large ViT-based model with not enough training data. This may harm the previous well-pretrained model because of the under-training. This can also be verified by the results of CLIP_L and CLIP_A, where we conduct PEFT approaches with training LoRA and adapters only. With these two approaches, the performance will get boosted compared with ViT_cen.
>
> We appreciate the reviewer's valuable comment. We will add the analysis above to the result analysis in Sec 4.2 and 4.3, respectively, in the final version of our paper.

---

### Decision · Program_Chairs · 2025-05-01

**Decision:**

Accept (poster)

**Comment:**

This paper proposes FedAG, a federated learning method to fine-tune large-scale vision-language models (e.g., CLIP) across distributed domains while preserving data privacy. FedAG introduces multiple domain-specific adapters to capture heterogeneous domain knowledge and maintain out-of-domain generalization. It happens in two-steps: Quality-Aware In-Domain Mutual Learning and Attention-Regularized Cross-Domain Learning. The former utilizes client models’ knowledge to refine the adapter for each domain, and the latter synthesizes logits from all adapters for inference on out-of-domain inputs. Experimental evaluations demonstrate that FedAG outperforms existing baselines in both in-domain accuracy and out-of-domain generalization.

Strengths:
1. The proposed method has practical value since it focuses on cross-silo federated learning and aligns with real-world privacy constraints.
2. The whole pipeline only trains very limited parameters, and the approach is scalable.
3. The reviewers highlight the quality of the empirical study and point out that the ablation studies are appropriate and informative.

Weaknesses:
1. Assumes a fixed number of domains; unclear how it handles dynamic or numerous clients.
2. Assumes high-quality and representative synthetic data, lacking in-depth analysis.
3. The main results do not sufficiently address the case in which domain information is not given

The authors address W1 satisfactorily and provided an intuitive modification to the approach; exploring this further is outside of the scope of this work.  Many weaknesses that were pointed out by reviewers where clarified by the authors in their rebuttal, namely, the comment about the this paper not taking into account data heterogeneity.  Regarding W3, the authors outline how they will address this in the camera ready, however, it is unclear if this will be sufficient.  Overall, the strengths outweigh the weaknesses, and the paper is suitable for ICML.